# Molecular Mechanisms of Estrogens in the Induction of Epithelial-to-Mesenchymal Transition and Metastasis in Breast Cancer

**DOI:** 10.3390/ijms26178589

**Published:** 2025-09-04

**Authors:** Javier Jiménez-Salazar, Luis Garcia-Melo, Nikola Batina, Adriana Alarcón-Aguilar, Armando Luna-López, Paulina Hernández-Garcés, Rebeca Damián-Ferrara, Pablo Damián-Matsumura

**Affiliations:** 1Department of Biology of Reproduction, Biological Sciences and Health Division, DCBS, Autonomous Metropolitan University (UAM), Av. Ferrocarril San Rafael Atlixco186, Col. Leyes de Reforma 1ª. Sec. Iztapalapa, Mexico City 09310, Mexico; 2Laboratory of Nanotechnology and Molecular Engineering, Electrochemistry Area, Chemistry Department, Basic and Engineering Division (DCBI), Autonomous Metropolitan University (UAM), Mexico City 09310, Mexico; gamlf@xanum.uam.mx (L.G.-M.); bani@xanum.uam.mx (N.B.); 3Department of Health Sciences, DCBS, Autonomous Metropolitan University (UAM), Mexico City 09310, Mexico; adyalagui@xanum.uam.mx; 4National Institute of Geriatrics, Secretaría de Salud (SSA), Mexico City 10200, Mexico; aluna@inger.gob.mx; 5Department of Electrical Engineering, Basic Sciences and Engineering Division (DCBI), Autonomous Metropolitan University (UAM), Mexico City 09310, Mexico; phg@xanum.uam.mx; 6Master’s in Pharmaceutical Design and Engineering Program, Technical University of Denmak, Anker Engelundsvej 1, Building 101A, DK-2800 Kgs., 2800 Lyngby, Denmark; rdferrara@outlook.com

**Keywords:** estrogen receptors, cell migration, cell adhesion, signal transduction, Src family kinases

## Abstract

Estrogens have been widely shown to induce cell proliferation in breast cancer (BC) cells. Recently, we have described their involvement in the induction of epithelial–mesenchymal transition (EMT), migration, and invasion. The aim of this work is to review the molecular mechanisms by which estradiol (E_2_) activates different signaling pathways, both genomic and non-genomic, through binding to different estrogen receptors (ERs), depending on the phosphorylated amino acid (Ser-118 or Tyr-537). The relevance of the present work lies in the molecular details of c-Src kinase activation by the membrane estrogen receptor (mER) and its effects on the early and late phases of EMT. This process initiates a loss of cell adhesion, leading to migration, which culminates in metastasis of cancer cells to distant tissues. Understanding how estrogens induce metastasis will facilitate the development of better strategies to counteract the lethality of BC. Finally, the quantification of Snail may serve as a molecular marker in the early stages of tumor progression, as well as the use of drugs against c-Src and ERs, as they may be therapeutic targets.

## 1. Introduction

Breast cancer (BC) is the most commonly diagnosed cancer worldwide and the fifth most common oncological cause of death [1]. Over 90% of breast cancer-related mortality is attributed to the metastasis of cancer cells to vital tissues, and the epithelial-to-mesenchymal transition (EMT) marks the beginning of this process [2]. Understanding the molecular mechanisms involved in the initiation of carcinogenesis could facilitate the development of early detection strategies and timely treatments, ultimately improving patient outcomes. In this review, we explore the various molecular mechanisms by which estradiol (E_2_), the most potent estrogen produced during a woman’s reproductive years, regulates EMT via different estrogen receptors (ERs). The genomic mechanisms of E_2_ are described by its interactions with the cytosolic and nuclear ER alpha and beta (ERα and ERβ, respectively), which induce transcription and protein synthesis that regulate different stages of EMT. We also detail the nongenomic pathways mediated by plasma membrane receptors, such as the membrane estrogen receptor (mER) and the G protein-coupled estrogen receptor (GPER).

In solid cancers, EMT is the process by which epithelial cells acquire a mesenchymal-like phenotype. This occurs as a result of the loss of epithelial characteristics and precedes the migration, invasion, and metastasis of tumor cells [3]. Traditionally, EMT has been considered a change between two cell populations: epithelial to mesenchymal. However, it is now known that precancerous epithelial cells gradually differentiate into partial states, which are characterized by the expression of specific markers and biochemical and phenotypic characteristics for each phase [4]. The reverse process of EMT is mesenchymal–epithelial transition (MET), which occurs after metastasis and gives rise to tumors in distant locations that have characteristics of the tissue from which they originated. Together, EMT and MET are known as epithelial–mesenchymal plasticity (EMP), a process that is imperative for cancer progression [5]. In the case of breast cancer, the elevated expression of EMP has been associated with worse patient outcomes, and inhibiting it has been proposed as a potential therapeutic strategy for blocking metastasis [6]. Not all tumor cells undergo EMP in the same way. This is because carcinoma cells lose their epithelial program in different ways, resulting in distinct modes of invasion and dissemination [7]. As will be seen later, estrogens induce EMT in BC cells. The genomic and non-genomic mechanisms by which E_2_ increases the expression of Snail, Twist, and Zeb1, transcription factors that are the main regulators of EMT, are detailed. On the other hand, E_2_ can reversibly repress the expression of proteins such as E-cadherin, ZO-1, and occludin, promoting migration, invasion, and metastasis in BC [8].

An interesting therapeutic approach is to suppress these factors to prevent EMT and BC recurrence; however, strategies against BC have not been entirely effective when the ER is inhibited to block EMT. This is due, in part, to the plasticity of tumor cells and the fact that E_2_ can stimulate or inhibit EMT depending on the context and tumor microenvironment, as will be explained later. In the case of BC, it has been observed that therapies targeting EMT molecular targets are only effective in the early stages, when ER dependence is present [9,10].

This review highlights the molecular details of the non-genomic mechanisms by which mER induces the activation of Rous sarcoma kinase (c-Src), a major promoter of EMT. Finally, we describe the signaling pathways through which genomic and non-genomic mechanisms converge to promote the disassembly of intercellular junctions, resulting in EMT, migration, and ultimately, the metastasis of BC cells.

## 2. Characteristics of the Estrogen Receptors

Estrogens act by binding to various intracellular or membrane receptors, activating genomic or nongenomic mechanisms. Additionally, they engage in crosstalk with other signaling pathways activated by peptide hormones or growth factors.

ERα belongs to the superfamily of nuclear receptors that function as transcription factors. It is encoded by the ESR1 gene, which is located on chromosome 6q25.1 [11]. ERα consists of 595 amino acids with an apparent molecular weight of 66 kilodaltons (kDa) and comprises three isoforms with molecular weights of 53, 46, and 36 kDa [12].

ERβ is encoded by the human gene ESR2, which is located on chromosome 14q22-24. Its protein is produced from eight exons, contains 530 aa, and has a molecular weight of 59.2 kDa. It exists in eight isoforms with approximate molecular weights of 57, 55, 54, 53, 53, 52, 48, 42, and 36 kDa (UniProtKB-Q92731; ESR2_HUMAN) [13,14]. Notably, the most studied isoform of ERβ is the one with an apparent molecular weight of 55.5 kDa [15]. Other types of ER that have been described are those that are anchored to the plasma membrane. These are known as mER and can be either ERα (mERα) or ERβ (mERβ) [16]. Immunodetection experiments using monoclonal antibodies directed against the amino- and carboxyl-terminal domains have shown that both the mERα (membrane-bound) and ERα (cytosolic/nuclear) proteins share the same molecular weight and electrophoretic mobility patterns [17]. The presence of a fatty acid chain facilitates membrane anchoring, leading the authors to conclude that ERα and mERα are the same protein [18].

ERα anchoring to the plasma membrane is accomplished via posttranslational modifications, such as palmitoylation or myristoylation. The amino acid sequence containing the serine residue (S447) enables S-palmitoylation, allowing the formation of a thioester bond with the ligand-binding domain of ERα. This domain resembles the domain homologous to the S-palmitoylated cysteine residue (C133) in human caveolin type 1, which promotes anchoring to the membrane [19].

The anchoring process is catalyzed by two different enzymes, DHHC-7 and DHHC-21, which are members of the protein acyl-transferase (PAT) family and contain an aspartate–histidine–histidine–cysteine domain [20]. The average PAT time for palmitate incorporation into ERα is approximately 7 min, whereas it is 60 min for ERβ, suggesting that ERα is a more efficient substrate for this process. The cyclic function of these enzymes induces the palmitoylation and depalmitoylation of ERα, affecting its activation and mobility across membrane microdomains. As a result, only a few ERα and ERβ proteins are membrane-localized and interact through caveolin type 1 palmitoylation [21].

When the E_2_ binds to the mERα, a process of slow depalmitoylation and dissociation from caveolin type 1 occurs, which enables its mobilization to other microdomains. [22]. The addition of fatty acids to ERα facilitates anchoring, increases hydrophobicity, and promotes mobility across the cell membrane. Evidence suggests that mER is dimerized in the presence of E_2_. In vitro and in breast cancer cells, it has been demonstrated that both mERα–mERα homodimers and mERα–mERβ heterodimers can be formed and are functional [23,24]. The E_2_ concentration must be high for the heterodimer to be achieved.

Under basal conditions, ERα is palmitoylated and associated with caveolin type 1. After stimulation with E_2_, ERα undergoes depalmitoylation, allowing for a conformational change that promotes homodimerization and subsequent association with other signaling proteins, thereby triggering different cellular functions. On the other hand, the formation of the mERα–mERβ heterodimer inhibits the interaction of the ERα complex with p85, thereby blocking the activation of the PI3-K pathway [25].

Notably, the amino acid sequence of mERα lacks cyclase, phospholipase, and kinase functions. Therefore, second messengers are synthesized in collaboration with other proteins, such as PKA and PI3-K, among others [26]. The signaling of mER occurs through physical interactions with molecules such as insulin-like growth factor receptor (IGFR), epidermal growth factor receptor (EGFR), small G protein Ras, and adaptor protein Shc (which bind directly to kinase-active receptors and activate MAPK signaling through the recruitment of Grb2/Sos) and the tyrosine kinase c-Src. Although ERα directly activates the PI-3K/Akt signaling pathway through its interaction with the regulatory protein subunit p85a, evidence suggests that Gαq or Gαs proteins may function as ER scaffolds in combination with certain effector kinases [27].

Finally, estrogens can bind to a third type of receptor, belonging to the family of 7-transmembrane domain receptors that act via G proteins, known as G protein-coupled receptor 30 (GPR30; UniProtKB-Q99527_GPER1_HUMAN). The gene encoding this receptor is located on chromosome 7p22.3, and the protein contains 375 aa, with an apparent molecular weight of 42.2 kDa [28]. It was originally cloned in 1992 and was later renamed G protein-coupled estrogen receptor (GPER) by the International Union of Basic and Clinical Pharmacology (IUPHAR). It was identified as an orphan receptor that has been associated with estrogen responses after its sequencing and identification in human cells [28].

## 3. Epithelial-to-Mesenchymal Transition (EMT) in Breast Cancer

The neoplastic cells of the mammary ductal epithelium, which are initially immobile, undergo structural changes that activate the expression of several genes associated with the mesenchymal phenotype. These changes allow the cells to dissociate from their neighboring cells, migrate, penetrate the basal membrane, and disseminate through lymphatic or blood vessels. Eventually, these cells implant in another organ to form a secondary tumor [29].

EMT, the process preceding metastasis, is characterized by epithelial cells losing their apical polarity and cubic structure. They acquire elongated shapes and develop a motility/migration front-to-back polarity, which is characteristic of mesenchymal cells. These changes favor the ability of cancer cells to penetrate the basement membrane to migrate and invade distant tissues [30].

For breast cancer cells to dissociate from neighboring cells, tight junctions (TJs) and adherens junctions (AJs) must be disassembled. As a result of the dissociation of these junctions, the loss of cell polarity is promoted, as apical proteins can migrate to basolateral areas and even translocate to the nucleus, causing tumor cells to lose their epithelial phenotype [29]. This occurs simultaneously with decreases in the expression of genes that encode proteins specific to epithelial cells, including E-cadherin, occludin, zonula occludens-1 (ZO-1), ZO-1-associated nucleic acid binding (ZONAB), metastasis-associated protein type 3 (MTA3), and claudins, among others. These changes favor the loss of intercellular junctions, intercellular adhesion, and epithelial detachment [31].

As EMT progresses, the expression of characteristic markers of mesenchymal tissue, including N-cadherin, vimentin, and transcription factors such as Snail, Slug, and Bmi1, increases. Like E-cadherin, N-cadherin is part of AJ junctions; however, the former is lost at the beginning of EMT and replaced by the latter. Vimentin is a structural protein that is a member of the intermediate filament family and is considered to be the major component of the cytoskeleton of mesenchymal cells [30]. The transcription factors Snail and Slug are members of the ZEB1 (zinc finger, E-box-binding homeobox) family and regulate the protein expression of E-cadherin and N-cadherin [32,33]. Bmi1 is a protein whose main function is to promote EMT via the transcriptional repression of E-cadherin, which favors cytoskeleton remodeling and enhances cell migration (Figure 1B).

To explain EMT more clearly, Tanaka and Ogishima divided the process into three sequential stages: (1) In the early stage, breast ductal cells have an epithelial phenotype (apical–basolateral), are attached to each other through cell junctions, and are anchored to the basement membrane by focal junctions. Some cells (orange) show changes in gene expression that favor the acquisition of the mesenchymal phenotype. (2) In the intermediate stage, the content and subcellular distribution of junctional complex proteins (AJs and TJs) decrease, favoring the loss of cell–cell and cell–basement membrane adhesion. The normal epithelial cell shape is misshapen (dysplasia), generating a mesenchymal phenotype (front–rear polarity) similar to that of fibroblasts. (3) In the late stage, epithelial markers are completely lost, and mesenchymal markers are expressed, allowing the cells to adopt a migratory phenotype characterized by remodeling of the cytoskeleton and degradation of the basement membrane, which enables them to migrate and invade other tissues (Figure 1A) [34].

In the early phase of EMT, ERα-bound E_2_ is able to repress the expression of E-cadherin, occludin, MTA-3, and ZO-1. It is also able to increase the expression of Snail. As explained later, the molecular mechanisms of E_2_ can be both genomic and nongenomic, as well as direct and indirect. Between the intermediate and late phases, E_2_ promotes the activation of mechanisms that are associated with the loss of cell–cell adhesion [34]. Notably, the ERα content decreases, in part, due to the increase in Snail, which transcriptionally represses it via a negative feedback mechanism. This causes no repression of Bmi1, suggesting that E-cadherin and occludin remain at low levels, independent of ERα activity. Notably, the E_2_–ERα complex induces N-cadherin synthesis through genomic mechanisms, starting at the early stage and reaching its maximum expression in the late stage of EMT (Figure 1B). N-cadherin is an important mesenchymal marker used in molecular classification [35].

Matrix metalloproteinases (MMPs) are zinc-dependent endopeptidases responsible for degrading the extracellular matrix (ECM). Their activation is necessary in the late phase of EMT, once the focal junctions have been dissociated. Tumor cells secrete MMP to degrade collagen, fibronectin, and laminins and can even break down substrates that are not part of the ECM, such as growth factors, integrins, and cadherins. This process allows cancerous epithelial cells to acquire mesenchymal characteristics to facilitate migration and invasion [36]. The main MMPs associated with migration and invasion in BC are MMP-1, 2, 9, 11, and 14 [37]. Various genomic and non-genomic mechanisms of estrogens have been described that regulate the expression and activation of these five MMPs [38]. An outstanding example of estrogen’s role in regulating TEM through MMPs is the loss of E-cadherin activity, which increases migration and metastasis in BC cells since it is cleaved by MMPs [39].

## 4. Genomic Mechanisms by Which E_2_ Induces EMT in Breast Cancer

The canonical genomic mechanism of action of E_2_ initiates when it diffuses across the plasma membrane due to its lipophilic nature and interacts with the intracellular ER (Figure 2, step 1). In their monomeric and cytoplasmic forms, ERs are associated with different heat shock proteins (HSPs), which maintain the ligand binding site in an “open state”, enabling access for E_2_ recognition [40]. Upon the binding of E_2_ to the ER, the interaction with HSPs is disrupted, which favors the phosphorylation of the ER at serine 118 (S^118^) [41]. The serine/threonine kinases that can carry out this process include Cdk7, Cdk2, and Erk, among others (Figure 2, step 2). This process is crucial for nuclear translocation; hence, the genomic mechanism takes place (Figure 2, step 3). When phosphorylation is carried out at tyrosine 537 (Y537), a nongenomic mechanism is activated [42], as explained below. Once in the nucleus, the ER binds directly to DNA at specific sequences known as estrogen response elements (EREs), which are located within the promoters of target genes (Figure 2, step 4). This allows for the transcriptional activation of these genes, which culminates in protein synthesis. The process is genomic and is called “canonical” because it was the first to be described. (Figure 2, step 5) [43].

The function of the ER can also occur through genomic mechanisms but is noncanonical. In these pathways, instead of binding directly to EREs, the ER forms complexes with other proteins, such as the transcription factors E26 transformation-specific sequence 1 (ETS-1), activator protein 1 (AP1; heterodimer Jun/Fos), specific protein types 1 to 3 (SP1–3), YY1 (transcription factor belonging to the GLI-Kruppel class of zinc finger proteins) or cAMP response element binding protein (CREB), among others. These interactions also facilitate the recruitment of transcriptional machinery and coregulators to activate the expression of target genes.

In breast cancer cells, E_2_ regulates the early stage of EMT by decreasing the cellular levels of E-cadherin, a transmembrane protein of the AJ. This regulation takes place via a canonical mechanism, since the E-cadherin gene has an ERE in the distal promoter, and the E_2_-ERα complex acts as a transcriptional repressor of E-cadherin through the recruitment of corepressors N-CoR and scaffold attachment factor B (SAFB1) to inhibit its transcription [44,45]. E_2_ can also negatively regulate E-cadherin expression through an indirect genomic mechanism mediated by the transcription factor E26 transformation-specific sequence 1 (ETS-1), where ERα functions as a transcriptional corepressor [46,47].

An important gene regulated by ERα is metastasis-associated protein 3 (MTA-3), which plays an important role in maintaining normal epithelial architecture by repressing the Snail 1 transcription factor and, consequently, regulating E-cadherin levels. ERα binds to SP1 response elements in the MTA3 promoter to increase its expression (Figure 1) [48,49].

Another mechanism regulating E-cadherin expression involves metastasis-associated protein 3 (MTA-3), which plays an important role in maintaining the normal architecture of the mammary epithelium. Its main function is the repression of the transcription factor Snail-1, which, in turn, inhibits the expression of E-cadherin [50]. The mechanism of action of the E_2_–ERα complex in the regulation of E-cadherin expression is genomic but indirect since it binds to the SP1 protein of the MTA-3 promoter SP1 site, in close proximity to a consensus ERE half-site, stimulating its synthesis [49,51]. Conversely, MTA-1 can inhibit ERα transcription [51,52]. In the early phase of EMT, the ER induces E-cadherin to increase through this mechanism; however, in the late phase, when the ERα concentration decreases, the opposite effect occurs, keeping E-cadherin concentrations low, which favors migration.

Estrogens can also decrease the expression of epithelial phenotype marker proteins that are part of intercellular junction complexes, such as occludin. The E_2_–ER complex indirectly inhibits occludin synthesis by acting through the transcription factors SP3 and YY1, since the gene encoding occludin lacks ERE [53].

Other epithelial markers that are regulated by estrogens include ZO-1 and ZONAB, which are part of the TJs. When disassembled, they can translocate to the nucleus, where the former functions as a transcription factor, and the latter is a coactivator of the former. Both proteins are negatively regulated by ERα through indirect mechanisms. ZO-1 expression is regulated when ERα binds to the Jun D protein, which is part of the AP-1 transcription factor, since an AP-1 response element is present in the proximal region of this gene. In the case of ZONAB, ERα binds to the CREB protein to inhibit its synthesis, since a cAMP response element (CRE) is present in the region close to the ZONAB promoter [54]. Estrogens regulate the expression of mesenchymal phenotype marker proteins, such as N-cadherin, vimentin, Snail, and SLUG, which are key promoters of EMT and cancer cell transformation. N-cadherin protein expression increases from the early stage of EMT, when it is induced by ERα, reaching its maximum in the late stage. This facilitates cell migration and cytoskeleton reorganization. The mechanism by which ERα increases its synthesis is of the canonical type, since the N-cadherin promoter presents ERE [55].

The transcription factor Snail is regulated by E_2_ through a genomic mechanism, as its promoter has a putative ERE located in its promoter [50]. On the other hand, Slug expression is decreased through ERα because the Slug promoter possesses three half-site EREs that are co-occupied by co-repressors [56]. SLUG is indirectly regulated by E_2_ since it lacks an ERE and is possibly regulated through potential SP-1, AP-1, or CRE sites that are present in the promoter, as described by a sequence homology search [57]. When Snail is overexpressed, the expression of E-cadherin, occludin, and even ERα decreases in the late stage of EMT. At this point, the participation of E_2_ is no longer required to conclude this process. A mechanism by which the E-cadherin concentration is low in the late phase of EMT is that the E_2_–ERα complex increases the expression of Snail. The overexpression of Snail decreases the expression of ERα, which functions as a repressor of Bmi1; therefore, an increase in Bmi1 decreases E-cadherin levels. Bmi1 negatively regulates E-cadherin expression by binding to a consensus E-box sequence present in its promoter region [58,59]. Recent studies confirmed the mechanism by which ERα regulates E-cadherin expression through silencing studies. This resulted in the increased synthesis of Bmi1 and, consequently, the transcriptional repression of E-cadherin, the loss of cell adhesion, and increased migration and metastasis [60]. On the other hand, E_2_ can bind to Erβ, which antagonizes the effects of ERα. The ERβ1 isoform inhibits EMT, migration, and metastasis by increasing E-cadherin expression through the activation of inhibitor of DNA-binding protein (Id1) in breast and ovarian cancer cells [61,62].

Vimentin expression in breast cancer cells may be regulated by E_2_; however, the mechanisms involved have not been fully elucidated. The vimentin promoter lacks ERE sites to which ERα can bind. Different sites, such as AP-1 in the distal promoter of vimentin and Sp1/3 in its proximal promoter, that could be used by ERα have been described, but this has yet to be demonstrated [63,64].

## 5. Non-Genomic Mechanisms of Estrogen on EMT in Breast Cancer

Estrogens also induce nongenomic mechanisms, characterized by their involvement of post-translationally modified proteins, resulting in rapid responses that occur within a few seconds to several hours. These processes are initiated at the cell membrane, where they activate signaling pathways either through transmembrane receptors (GPERs) or through intracellular receptors anchored to the inner side of the cell membrane [65]. Estrogen-induced nongenomic mechanisms indicate that different phosphorylation sites of ERα are critical for hormone sensitivity, nuclear localization, DNA binding, transcription activation, interaction with other proteins, and association with the cell membrane [45]. In particular, serine residues 118 and 167 (S118 and S167, respectively) play significant roles in activating nuclear coregulators, binding to EREs, and amplifying the genomic response [66]. On the other hand, the phosphorylation of tyrosine residue 537 (Y537) of ERα facilitates its interaction with the SH2 domain of Rous sarcoma kinase (c-Src) and favors signal transduction from the cell membrane [67].

ERα-c-Src signaling has been widely described as a nongenomic pathway, and the mechanism is detailed in Figure 3. ERα activation by E_2_ binding induces conformational changes in Erα, allowing its phosphorylation at residue Y537 by various kinases, including c-Src itself [66]. The phosphorylation of Y537 facilitates the anchoring of ERα to the cytoplasmic membrane via the addition of a fatty acid residue (palmitoylation or myristoylation). This posttranslational modification gives rise to what is known as the membrane estrogen receptor (mER). The c-Src kinase presents three domains that regulate its activity: the c-Src homology domains type 2 and type 3 (SH2, SH3), whose functions are the recognition of tyrosine-phosphorylated residues and proline-rich sequences, respectively, and the protein tyrosine kinase (PTK) domain. In addition, c-Src has two phosphorylation residues: Y527 and Y416. The former maintains the enzyme in its inactive or closed conformation, since the catalytic domain of PTK is hidden, in addition to enabling interaction with mERα. This binding favors the conformational change of c-Src and the exposure of the residue at Y416 that can be phosphorylated [68] while dephosphorylating Y525, thereby generating a functional form of c-Src.

The molecular mechanism by which mERα interacts with c-Src is mediated by the MEMO protein (mediator of ErbB2-driven cell motility protein) [69]. Frei et al. suggested that the cytoplasmic interaction of ERα and c-Src is mediated by MEMO, which enhances E_2_-induced phosphorylation at residues Y416 of c-Src and Y537 of ERα. The formation of the MEMO, c-Src, and ERα complex induces the activation of c-Src, which subsequently phosphorylates the Y537 residue of ERα, stimulating the formation of the ERα/c-Src complex and preventing ERα nuclear translocation [70].

The role of MEMO as a key regulator of nongenomic pathways that stimulate estrogen-induced migration and proliferation in cancer cells has been highlighted. MEMO can be present in both the cytoplasm and nucleus, suggesting that it may act on different proteins. This is important because it is overexpressed in more than 40% of primary breast cancer tumors, so its abundance in the cytoplasm is related to the aggressiveness of the disease, including luminal subtype B breast cancer, recurrence, and increased mortality [71].

As mentioned, at the initial stage of EMT, changes in cell polarity occur due to alterations in intercellular junction complexes (AJ, TJ, and focal junctions [cell–basal membrane]). Our research revealed that activation of the ERα–Src pathway increases the expression of N-cadherin and Snail; decreases the expression of the E-cadherin, occludin, and CRB-3 proteins; and changes the localization of ZO-1 and ZONAB from the TJs to the nucleus, thereby increasing paracellular permeability, promoting the loss of adhesion, and enhancing migration and invasion [54].

One of the nongenomic mechanisms, as described in Figure 4, indicates that the interaction of E_2_ with ERα favors its phosphorylation at the Y537 residue. This change enables the palmitoylation or myristoylation of the receptor, facilitating its anchoring to the membrane. Once mERα is active, it stimulates various signaling pathways through interactions with c-Src kinase. The interaction between mERα and c-Src is mediated by the SH2 domain of c-Src, which recognizes the Y537 residue of mERα, favoring the full activation of c-Src through the autophosphorylation of the Y416 residue. Once p-Src is activated (p-Src Y416), it binds ZO-1, forming the p-Src-Y416-ZO-1 complex. This complex induces the dissociation of the ZO-1 and ZONAB proteins from the junction complex and facilitates their translocation to the nucleus, where they function as a coactivator (ZO-1) and a transcription factor (ZONAB) for genes involved in proliferation and survival in breast cancer, such as human epidermal growth factor 2 (HER2). Additionally, c-Src kinase forms a p-Src-Y416/E-cadherin complex that promotes the delocalization of occludin from TJs and E-cadherin from AJs. On the other hand, the E_2_/ERα complex increases the expression of the mesenchymal markers N-cadherin, vimentin, and Snail through genomic mechanisms in the late phase of EMT. These changes allow migration and invasion to take place in BC cells.

## 6. Insights into the Molecular Mechanisms of Estrogens and the Regulation of EMT in Clinical Practice

The abnormal reactivation of EMT in BC cancer cells increases their migration, invasiveness, and ultimately, metastasis. We have demonstrated that one mechanism associated with EMT induction is mediated by E2 via a non-genomic mechanism that is dependent on c-Src kinase activation. This allows tumor cells to grow uncontrollably and become more resistant to chemotherapy and immunotherapy [31,72]. The information presented in this review suggests the possibility of searching for new molecular markers that can be detected in the early or intermediate stages of breast cancer (BC), as well as novel therapeutic targets based on the molecular mechanisms of action of estrogen (E2).

In the clinic, the quantification of E-cadherin protein has been used as a predictor of poorer prognosis because its decrease has been associated with the progression of EMT. However, reports indicate that E-cadherin promotes the development of metastasis and invasion in BC when aberrantly high expression is present. For this reason, this molecular marker is not fully reliable. We propose the use of Snail instead of E-cadherin, since its expression increases at the early phase of EMT, reaching its maximum in the late phase. Snail protein has the ability to negatively regulate the expression of E-cadherin in the early phase of EMT and increase that of N-cadherin in the late phase of EMT. Because of these characteristics, we propose that high levels of Snail can serve as a predictive marker in the different phases of EMT in BC.

In addition, some clinical research indicates that BC patients with high expression of genes associated with middle EMT (EPCAM, CD106+, and ALDH1) exhibited more metastases and resistance to treatment compared to patients who expressed genes associated with early EMT (CD24 and CD44). Based on these observations and the findings presented in this review, we propose using E-cadherin, occludin, and ERα as markers of a good prognosis, taking the aforementioned considerations into account. Conversely, the detection of ZO-1, ZONAB, N-cadherin, and Snail proteins may serve as markers of poor prognosis. The former two have the ability to increase HER-2 oncogene expression, and Snail promotes migration and invasion. Additionally, Snail is a transcriptional repressor of ERα (Figure 1).

Another important aspect is the relationship between neoadjuvant chemotherapy (NAC) and EMT in ER-positive breast cancer, where estrogen plays a very important role. In patients with BC, studies have shown that a lower response to NAC (using a sequential treatment of anthracyclines followed by taxanes) is associated with changes in the expression of genes associated with EMT detected in diagnostic biopsies obtained in early stages [73]. This can be explained by the fact that high ER expression decreases the expression of epithelial markers and increases the expression of mesenchymal markers, which is associated with chemoresistance. Additionally, cells resistant to NAC can continue to proliferate because estrogens activate DNA repair mechanisms and inhibit apoptosis, both of which are associated with estrogen-mediated chemoresistance [72].

## 7. Conclusions

Estrogens are widely associated with cell proliferation in both normal breast tissue and cancer cells. Recently, they have been described as inducers of processes such as EMT, migration, invasion, and metastasis. In this review, we detail the molecular mechanisms by which E_2_ can regulate the different phases of EMT through both genomic and nongenomic mechanisms via the different ERs involved in this process. We describe a novel molecular mechanism by which E_2_ interacts with c-Src, allowing for its activation and the induction of signaling at the plasma membrane level, which regulates the expression and localization of both epithelial phenotype marker proteins (ZO-1, ZONAB, occludin, and E-cadherin) and mesenchymal markers (N-cadherin, Snail, and vimentin).

This evidence provides the basis for the use of c-Src inhibitors, which are currently available (Dasatinib^®^), in combination with an agent that blocks ERα function (faslodex/fulvestrant^®^), as an alternative in the treatment of early-stage BC, with the aim of halting cell progression in the EMT and preventing metastases from forming. Combining Src inhibitors with endocrine therapies can be more effective than using either treatment alone because it can help overcome or delay resistance to endocrine therapies and enhance the efficacy of Src inhibitors.

On the other hand, the use of new markers of tumor progression based on their molecular mechanisms of action is proposed. In the clinic, the quantification of E-cadherin protein has been used as a predictor of poorer prognosis because its decrease has been associated with the progression of EMT. However, reports indicate that E-cadherin promotes the development of metastasis and invasion in BC when aberrantly high expression is present. For this reason, this molecular marker is not fully reliable. We propose the use of Snail instead of E-cadherin, since its expression increases at the early phase of EMT, reaching its maximum in the late phase. Snail protein has the ability to negatively regulate the expression of E-cadherin in the early phase and increase that of N-cadherin in the late phase of EMT. Because of these characteristics, we propose that high levels of Snail can serve as a predictive marker throughout the different phases of EMT in BC.

Overall, the aforementioned research suggests that estrogens and their genomic and non-genomic signaling pathways contribute to tumor chemoresistance. Improving our understanding of the molecular mechanisms of estrogen-mediated DNA repair and EMT induction could lead to the development of novel combination therapies or tumor markers.

## Figures and Tables

**Figure 1 ijms-26-08589-f001:**
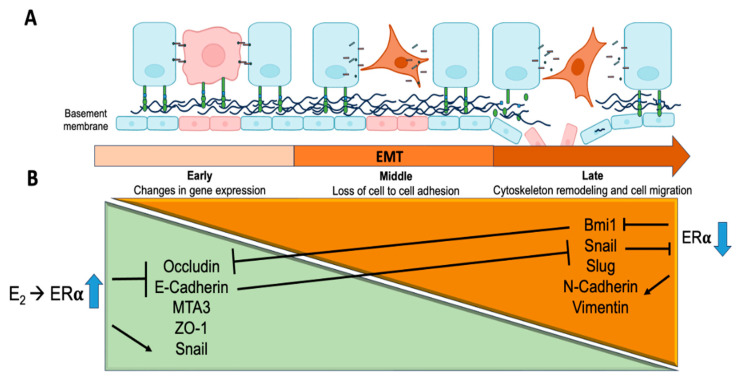
EMT presents three phases or stages: early, middle, and late. (**A**) First, the epithelial cells of mammary ducts exhibit changes in gene expression that cause the loss of polarity and, therefore, the epithelial phenotype (orange-colored cells). In the intermediate stage, cells acquire the mesenchymal phenotype characterized by changes in their structure, weakening the cell–cell junction and the transition of the cell to the basement membrane (cherry-colored structures). Finally, cancer cells undergo remodeling of the cytoskeleton, which allows them to migrate and invade other tissues. (**B**) In the early stage of EMT, the expression of ERα is high, and, in the presence of E_2_, it stimulates the synthesis of Snail and decreases the expression of the epithelial markers occludin, E-cadherin, MTA3, and ZO-1. In the late phase, the synthesis of mesenchymal markers such as Bmi1, Snail, Slug, N-cadherin, and vimentin is increased. Higher levels of the transcription factor Snail induce a lower expression of ERα, and as a consequence, the expression of Bmi1, a transcriptional repressor of E-cadherin, which favors tumor progression, is increased.

**Figure 2 ijms-26-08589-f002:**
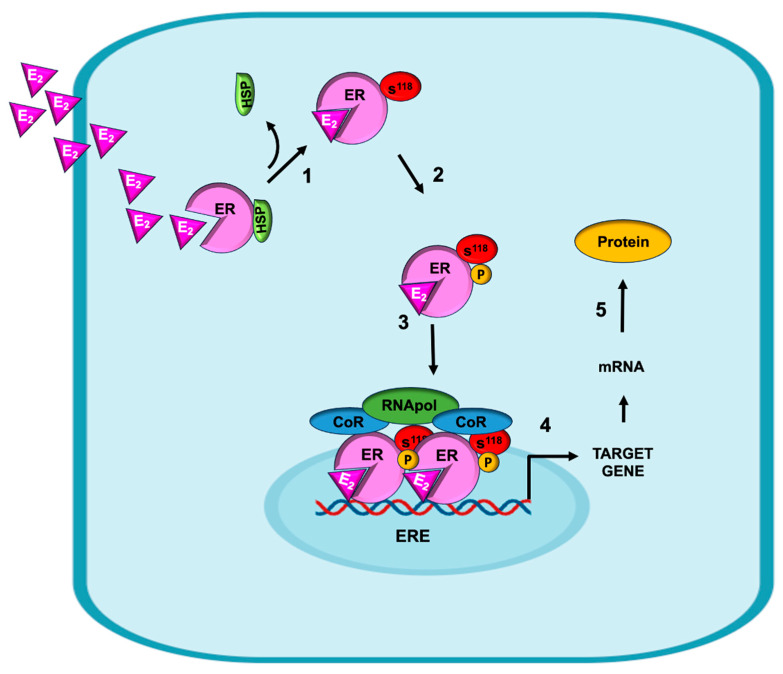
Canonical mechanism of estrogens. Estradiol (E_2_) diffuses through the plasma membrane and binds to the ER, and the ligand–receptor interaction generates conformational changes in the receptor that allow it to dissociate from heat shock proteins (HSPs) and favor their phosphorylation. The ER subsequently translocates to the nucleus, where it forms dimers and binds to specific sequences in the DNA known as estrogen response elements (EREs) present in target genes, favoring the transcription and subsequent translation of estrogen-responsive proteins.

**Figure 3 ijms-26-08589-f003:**
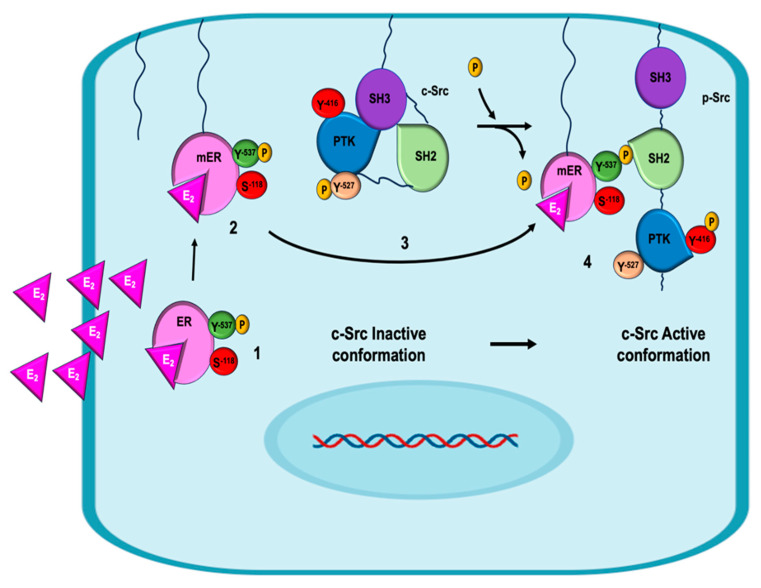
The interaction of E_2_ with the ER favors its phosphorylation at the Y537 residue instead of S118 [step 1]. This phosphorylation facilitates the anchoring of the ER to the cytoplasmic membrane via the addition of a fatty acid residue (palmitoylation or myristoylation), which converts cytoplasmic ERα into the membrane ERα (mER) [step 2]. On the other hand, the c-Src kinase presents an inactive conformation because it is phosphorylated at the Y527 residue, which keeps it closed without exposing the catalytic domain [step 3]. The E_2_/mERα complex (phosphorylated at Y537) is able to bind to the SH2 domain of c-Src in its inactive conformation and render it active by causing the loss of Y527 phosphate and favoring phosphorylation at Y416. This causes c-Src to present its active conformation, exposing the catalytic site (PTK) [step 4].

**Figure 4 ijms-26-08589-f004:**
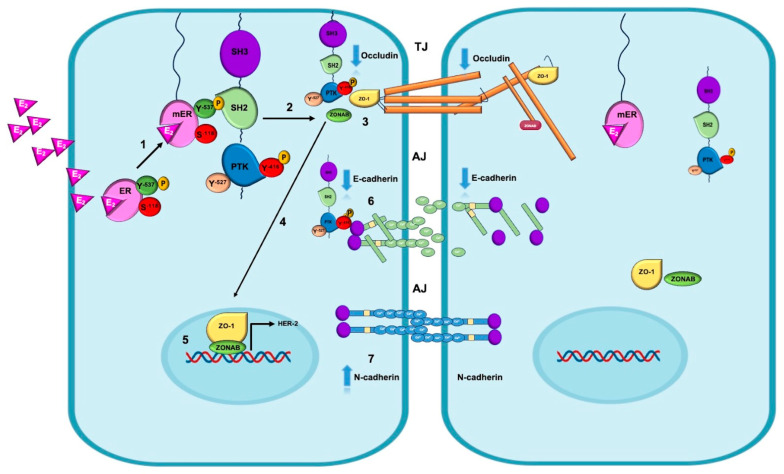
The interaction of E_2_ enables ERα phosphorylation at the Y537 residue, anchoring it to the plasma membrane and the formation of mERα [step 1]. The interaction of the E_2_/mERα complex with c-Src promotes its activation via phosphorylation of the Y416 residue (p-Src Y416) [step 2], which interacts with the proteins that compose tight junctions (TJs). Then, the p-Src/Y416/ZO-1 complex is formed, promoting the downregulation of occludin and the dissociation of the ZO-1 and ZONAB proteins from TJs [step 3]. This favors the translocation of both proteins to the nucleus [step 4], as they function as a coactivator and a transcription factor, respectively, activating the expression of the HER2 oncogene [step 5]. Moreover, active c-Src kinase can form the p-Src/Y416/E-cadherin complex, inducing its phosphorylation and delocalization at adherens junctions (AJs) and promoting the downregulation of this epithelial marker [step 6]. The E_2_/ERα complex is able to increase the expression of the mesenchymal markers N-cadherin, vimentin, and Snail through genomic mechanisms in the late phase of EMT [step 7]. These changes allow migration and invasion to take place in BC cells.

## Data Availability

Not applicable.

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
