# Peer review of "Molecular Mechanisms of Estrogens in the Induction of Epithelial-to-Mesenchymal Transition and Metastasis in Breast Cancer"

_ijms, 2025, doi:10.3390/ijms26178589_

Round 1
Reviewer 1 Report
Comments and Suggestions for Authors
In this review, the authors aim to provide an overview of the molecular mechanisms by which estrogens contribute to the induction of epithelial-to-mesenchymal transition (EMT) and metastasis in breast cancer. While the topic is of interest and relevance to the field, the manuscript requires major revisions before it can be considered for publication. Below are specific suggestions to improve the clarity and accuracy of the work:
- The statement at line 37-38 "over 90% of breast cancer-related mortality is attributed to metastasis due to estrogen receptors" is inaccurate and oversimplified. While metastasis is indeed the leading cause of breast cancer mortality, it is not solely related to estrogen receptor (ER) signaling. Multiple mechanisms, including ER-independent pathways, are involved. Moreover, metastatic progression is not limited to ER-positive breast cancer; in fact, some ER-negative subtypes, such as triple-negative breast cancer (TNBC), are associated with higher metastatic potential. This sentence should be revised to reflect the broader and more complex nature of metastasis in breast cancer.
- The concept of epithelial–mesenchymal transition (EMT) should be introduced with greater depth and accuracy in the introduction.
- The authors should also discuss the mesenchymal-to-epithelial transition (MET) as a complementary and reversible process to EMT. This plasticity plays a critical role in metastatic colonization and may be influenced by estrogen signaling.
- In several instances, the review references estrogen-related mechanisms based on studies conducted in non-breast cancer contexts. The authors should clearly specify when the evidence is derived from other pathologies and, where possible, focus on breast cancer-specific mechanisms. This distinction is important, as estrogen signaling pathways can vary significantly depending on tissue and disease context.
- The figures included in the manuscript are unclear and appear to be cropped. All visual materials should be revised.
- The manuscript would benefit from a more comprehensive discussion of how the molecular mechanisms described could translate into clinical practice. Specifically: How does current breast cancer therapy, particularly endocrine therapy, affect estrogen signaling and EMT? What is the relationship between resistance to anti-estrogen therapies and EMT/metastatic progression? Addressing these questions would enhance the translational relevance of the review.
- Several minor errors should be corrected: Line 150: "TEM" should be corrected to "EMT", Line 218: "RE" should be corrected to "ER"
Author Response
Comment 1: The statement at line 37-38 "over 90% of breast cancer-related mortality is attributed to metastasis due to estrogen receptors" is inaccurate and oversimplified. While metastasis is indeed the leading cause of breast cancer mortality, it is not solely related to estrogen receptor (ER) signaling. Multiple mechanisms, including ER-independent pathways, are involved. Moreover, metastatic progression is not limited to ER-positive breast cancer; in fact, some ER-negative subtypes, such as triple-negative breast cancer (TNBC), are associated with higher metastatic potential. This sentence should be revised to reflect the broader and more complex nature of metastasis in breast cancer.
Response 1: Thank you for pointing this out. We agree that the sentence "over 90% of breast cancer-related mortality is attributed to metastasis due to estrogen receptors" is incorrect. Therefore, we revised and corrected this sentence (Lines 38-40, page 2).
Comment 2: The concept of epithelial–mesenchymal transition (EMT) should be introduced with greater depth and accuracy in the introduction.
Response 2: We agree with this and have incorporated your suggestion in lines 51-54, page 2.
Comment 3: The authors should also discuss the mesenchymal-to-epithelial transition (MET) as a complementary and reversible process to EMT. This plasticity plays a critical role in metastatic colonization and may be influenced by estrogen signaling.
Response 3: Agree. Your suggestion has been incorporated in lines 54-62, pages 2-3.
Comment 4: In several instances, the review references estrogen-related mechanisms based on studies conducted in non-breast cancer contexts. The authors should clearly specify when the evidence is derived from other pathologies and, where possible, focus on breast cancer-specific mechanisms. This distinction is important, as estrogen signaling pathways can vary significantly depending on tissue and disease context.
Response 4: Agree. Where possible, references to estrogen-related mechanisms reported in breast cancer cells or patients were changed. In other paragraphs, specific diseases were identified.
Comment 5: The figures included in the manuscript are unclear and appear to be cropped. All visual materials should be revised.
Response 5: Agree. All figures (4) were revised and modified.
Comment 6: The manuscript would benefit from a more comprehensive discussion of how the molecular mechanisms described could translate into clinical practice. Specifically: How does current breast cancer therapy, particularly endocrine therapy, affect estrogen signaling and EMT? What is the relationship between resistance to anti-estrogen therapies and EMT/metastatic progression? Addressing these questions would enhance the translational relevance of the review.
Response 6: We agree with this and have incorporated your suggestion at the end of “Conclusion”, in lines 367-375, page 15.
Comment 7: Several minor errors should be corrected: Line 150: "TEM" should be corrected to "EMT", Line 218: "RE" should be corrected to "ER".
Response 7: Agree. All minor errors were corrected, including “TEM” corrected to “EMT” and “RE” to “ER”.
Reviewer 2 Report
Comments and Suggestions for Authors
Comment 1: Line 73: Authors have written 'ERα and mERα ' however I assume they mean 'ERα and mER'.
Comment 2: Line 75: Authors have written 'ERα anchoring from ER to plasma mebrane' but ERα is a nuclear receptor how it binds to ER and plasma membrane? Please restructure this sentence.
Comment 3: Line 89: Authors are mentioning E2 here for the first time, it will be great to have at least one short sentence for E2's introduction.
Comment 4: Line 107: Authors have swapped 'ERα' to 'REα' at many places. Is this a typo?
Comment 5: Authors have written 'TEM' do they mean 'EMT' here?
Comment 6: While discussing EMT in breast cancer, authors should mention that along with cytoskeletal remodifications, cancer cells express matrix degrading enzymes, another hallmark for EMT. This part is excluded in the manuscript.
Comment 7: Figure 1: 'cell adhession' : correct it. 'Cytoskeleton remodelation': correct it.
Comments on the Quality of English LanguageThe overall quality is satisfactory; however, the manuscript requires proofreading, and certain sentences should be revised to enhance scientific clarity
Author Response
Comment 1: Line 73: Authors have written 'ERα and mERα ' however I assume they mean 'ERα and mER'.
Comment 2: Line 75: Authors have written 'ERα anchoring from ER to plasma membrane' but ERα is a nuclear receptor how it binds to ER and plasma membrane? Please restructure this sentence.
Response 1 and 2: Thank you for pointing this out. The paragraphs were revised and modified to clearly indicate that ‘ERα’ anchors to the plasma membrane and becomes ‘mERα’. The restructured sentences are in lines 90-92, page 4. The manuscript thoroughly describes the molecular mechanism by which ERα is determined to remain in the cytoplasm or anchor to the membrane.
Comment 3: Line 89: Authors are mentioning E2 here for the first time, it will be great to have at least one short sentence for E2's introduction.
Response 3: Agree. A sentence that introduces E2 has been incorporated in lines 42-44, page 1.
Comment 4: Line 107: Authors have swapped 'ERα' to 'REα' at many places. Is this a typo?
Comment 5: Authors have written 'TEM' do they mean 'EMT' here?
Response 4 and 5: Thank you for pointing out these errors. The entire text has been reviewed, and the typos have been corrected.
Comment 6: While discussing EMT in breast cancer, authors should mention that along with cytoskeletal remodifications, cancer cells express matrix degrading enzymes, another hallmark for EMT. This part is excluded in the manuscript.
Response 6: We agree with this and have included your suggestion in lines 187-196, page 6.
Comment on the Quality of English Language: The overall quality is satisfactory; however, the manuscript requires proofreading, and certain sentences should be revised to enhance scientific clarity.
Response: The manuscript was sent out to be proofread for English language quality, with a particular focus on ensuring scientific clarity.
Round 2
Reviewer 1 Report
Comments and Suggestions for Authors
The authors have addressed most of the concerns raised, and the manuscript shows clear improvement compared to the initial submission.
However, I still have two minor concerns:
- I recommend including a paragraph that discusses how the molecular mechanisms described can be translated into clinical practice. Some of these points are already mentioned in the conclusion (lines 427–432 and 445–459), but expanding them into a dedicated section would allow the authors to better highlight the therapeutic aspects and their connection to the molecular pathways presented. This would also enable the conclusion to be more concise and refined.
- The current formatting of the manuscript resembles that of a research article rather than a review. A dedicated "Results" section is not appropriate, as the manuscript does not present original data.
Author Response
Comment 1: I recommend including a paragraph that discusses how the molecular mechanisms described can be translated into clinical practice. Some of these points are already mentioned in the conclusion (lines 427–432 and 445–459), but expanding them into a dedicated section would allow the authors to better highlight the therapeutic aspects and their connection to the molecular pathways presented. This would also enable the conclusion to be more concise and refined.
Response 1: We appreciate the time and effort the reviewer dedicated to providing valuable feedback on the manuscript. The section “Insights into the Molecular Mechanisms of Estrogens and the Regulation of EMT in Clinical Practice” was included, discussing how the molecular mechanisms of estradiol regulate EMT and how this translates into clinical practice. As suggested, the information was expanded, highlighting certain therapeutic aspects and their connection to the molecular pathways presented. The “Conclusion” was made more concise and refined.
Comment 2: The current formatting of the manuscript resembles that of a research article rather than a review. A dedicated "Results" section is not appropriate, as the manuscript does not present original data.
Response 2: The manuscript does not include the “Results” section. There may have been some confusion with another document.